# Effects of Glucose Metabolism, Lipid Metabolism, and Glutamine Metabolism on Tumor Microenvironment and Clinical Implications

**DOI:** 10.3390/biom12040580

**Published:** 2022-04-14

**Authors:** Longfei Zhu, Xuanyu Zhu, Yan Wu

**Affiliations:** 1The Fourth Affiliated Hospital of Jiangsu University, Zhenjiang 212001, China; 3181401051@stmail.ujs.edu.cn (L.Z.); 3181401052@stmail.ujs.edu.cn (X.Z.); 2School of Medicine, Jiangsu University, Zhenjiang 212001, China

**Keywords:** metabolism, tumor microenvironment, immunotherapy, glycolysis, fatty acid metabolism, glutaminolysis

## Abstract

In recent years, an increasingly more in depth understanding of tumor metabolism in tumorigenesis, tumor growth, metastasis, and prognosis has been achieved. The broad heterogeneity in tumor tissue is the critical factor affecting the outcome of tumor treatment. Metabolic heterogeneity is not only found in tumor cells but also in their surrounding immune and stromal cells; for example, many suppressor cells, such as tumor-associated macrophages (TAMs), myeloid-derived suppressor cells (MDSCs), and tumor-associated T-lymphocytes. Abnormalities in metabolism often lead to short survival or resistance to antitumor therapy, e.g., chemotherapy, radiotherapy, targeted therapy, and immunotherapy. Using the metabolic characteristics of the tumor microenvironment to identify and treat cancer has become a great research hotspot. This review systematically addresses the impacts of metabolism on tumor cells and effector cells and represents recent research advances of metabolic effects on other cells in the tumor microenvironment. Finally, we introduce some applications of metabolic features in clinical oncology.

## 1. Introduction

It is difficult to use a uniform approach for the treatment of tumors due to their considerably different origins, tissue types, and genotypes. This is manifested not only in the heterogeneity of cell type, including tumor cells, immune infiltrating cells, and stromal cells, which shape the microenvironment of the area where malignant tumor cells are located; cell numbers; and cell composition, but also in the heterogeneity of cellular metabolism [1]. In the early 1920s, the Warburg effect was first described by Otto Heinrich Warburg [2]. In his experiments, he reported that even in the presence of oxygen, tumor cells tend to carry out glycolysis to provide energy for themselves [3]. However, he attributed this phenomenon to the impairment of mitochondrial function, which was later proved to be incorrect [4]. This was the first discovery of the reprogramming process of sugar metabolism in human tumor cells. Thirty years later, researchers have discovered the importance of glutamine metabolism in HeLa cells [5]. Since then, the impact of immune metabolism on the tumor microenvironment has been in progress. In 2011, Hanahan et al. [6] suggested that metabolic abnormalities might be an important marker for predicting cancer development [6]. In addition, they also emphasized the critical role of metabolic alterations in immune escape, invasion, and metastasis of tumor cells.

The tumor microenvironment is an emerging concept that encompasses the stromal components and cells surrounding the tumor tissue, including fibroblasts, mesenchymal stem cells, adipocytes, peritoneal cells, endothelial cells and immune cells [7]. Reprogramming of tumor microenvironment profoundly affects the outcome of tumor cells to chemotherapy, radiotherapy, targeted therapy, and immune checkpoint therapy, which is itself regulated by metabolism (e.g., glycolysis, glutaminolysis, fatty acid metabolism) [8]. The importance of metabolic effects on the tumor microenvironment has been gradually recognized with the advancement of metabolic studies; therefore, changes in targeted metabolism are of great importance for cancer treatment. For example, CB839, a glutaminase (GLS) inhibitor, not only has antitumor capabilities [9] but has also been shown to enhance the therapeutic effect of chimeric antigen receptor T cell (CAR-T) [10]. Broadly speaking, inhibition of anyone pathways in glucose metabolism and glutamine metabolism in tumor cells in combination with immunotherapy increases the sensitivity of tumor cells to immunotherapy. Furthermore, previous studies have focused on the metabolism of tumor cells themselves but neglected the immune cells that play an important role in the microenvironment, while the metabolism of immune cells does play an important role in the function of immune cells themselves and in the process of tumor growth [11].

Of course, there are some limitations in this approach at present, such as the mechanism of certain metabolic pathways and their effect on the tumor microenvironment, which still need to be further explored, as well as the selection of targeted metabolic drugs, which needs to be more emphasized for target specificity to tumor cells [12]. Research on the interconnection of metabolism and immune escape in oncology is emerging, and the detailed in-depth mechanisms, as well as drugs, will be explored in the future. However, previous studies have undoubtedly contributed to the further understanding of the significance of metabolism and immune escape.

## 2. Effect of Metabolism on Cancer Cells

### 2.1. Effect of Glucose Metabolism on Tumor Cells

According to the Pasteur effect, in the absence of oxygen, glucose is oxidized by cells to lactate rather than entering the tricarboxylic acid (TCA) cycle; however, in some tumor cells, even in the existence of oxygen, glucose enters the cytoplasm and undergoes a chemical reaction catalyzed by pyruvate kinase and lactate dehydrogenase to produce lactate and nicotinamide adenine dinucleotide ^+^ (NAD^+^) [3,13]. One molecule of glucose through the TCA cycle produces nearly 20 times as much energy as glycolysis [14]. However, one molecule of glucose can produce 10–100 times more adenosine triphosphate (ATP) per unit time by glycolysis compared to the TCA cycle in mitochondria [15]. One explanation for why tumor cells would choose such a productive method is that glycolysis helps tumor cells to better compete for resources in a nutrient-limited micro-environment [15]. Furthermore, Chang et al. confirmed that tumor cells compete with T-lymphocytes for glucose in the microenvironment, suppressing antitumor immunity and indirectly promoting the growth of cancer cells [16]. Interestingly, lactate, as the product of glycolysis, has been also found to regulate the tumor cells. By using ^13^C isotope labeling, studies have documented that approximately 50% of the carbon in lactate was transferred to lipids in HeLa and H460 cells, and moreover, they also proposed that mitochondria-associated lactate dehydrogenase converts lactate to pyruvate [17]. That is, lactate may serve as a fuel for the TCA cycle. Moreover, lactate can also lead to histone lactylation that convert type 1 tumor-associated macrophages (TAMs) with anti-inflammatory effects into type 2 TAMs with pro-inflammatory effects. M1-like TAMs are more inflammatory and use glycolysis, fatty acid synthesis (FAS), and amino acid metabolism to provide nutrients [18]. Conversely, M2-like TAMs exhibit a more suppressive phenotype that utilizes the TCA cycle and fatty acid oxidation (FAO) [18]. The results of a later study in patients with non-small cell lung cancer injected with lactate and using an isotopic tracer method reinforced these ideas [19].

At the same time, there is also an alternative branch of glycolysis to produce the pentose, which is usually named the pentose phosphate pathway (PPP). Cells do not produce or consume energy in the form of ATP in this pathway. The glucose-6-phosphate (G-6-P) produced by glycolysis is catalyzed by G-6-P dehydrogenase (G6PDH) and 6-phosphogluconate dehydrogenase (6PGDH) to produce ribose-5-phosphate (R-5-P) and nicotinamide adenine dinucleotide phosphate (NADPH) [20]. R-5-P can re-enter the progress of glycolysis by transforming into fructose-6-phosphate (F-6-P) and glyceraldehyde-3-phosphate (G-3-P) (Figure 1). NADPH is one of the most important antioxidants in the body, transferring electrons and hydrogen displaced by the energy of sunlight, reducing intracellular oxidation by regulating the conversion of oxidized glutathione to reduced glutathione, working with a wide variety of enzymes, and is considered one of the universal electron carriers [21]. Some intracellular metabolic processes produce oxides such as reactive oxygen species (ROS) that bind to proteins, and the presence of NADPH will reduce these proteins and maintain their function [21]. Meanwhile, for cancer cells, R-5-P produced by PPP can provide a high rate of nucleic acid synthesis [21]. Therefore, the PPP with high activation in cancer cells not only reduces the damage to cancer cells by reactive ROS but also provides material for DNA synthesis [22]. PPP is regulated by many factors; when in a homeostatic environment, it is regulated by key enzymes, viz., G6PDH, which regulates the production of R-5-P and NADPH, and when in an environment where homeostasis is dysregulated, it is regulated by immunosuppressive factors, such as transforming growth factor-β (TGF-β). Therefore, a better understanding of the connections and differences between the glucose glycolytic pathway and PPP pathway in tumor cells is important for elucidating the metabolic reprogramming of tumor cells, which is associated matter to revealing the mechanism of immune escape, drug resistance, and therapy failure. Further studies of abnormalities in the PPP metabolic pathway may discover new therapeutic targets, which can be combined with existing treatments to improve treatment.

### 2.2. Effect of Glutamine Catabolism on Tumor Cells

Since the discovery of Warburg effect, glucose has been considered as a core molecule of tumor metabolism research. The mechanism of its effect on tumor cells and adjacent cells has also been investigated in depth by researchers; in particular, the state-of-the-art inventions and use of instruments as well as technologies have further revealed their internal mechanisms. In recent decades, scientists have also become more concerned about the effects of amino acid metabolism in the tumor microenvironment, particularly the breakdown of glutamine, which is one of the most thoroughly studied amino acids [5].

Glutamine is an important metabolic fuel that helps rapidly proliferating cells to meet the growing demand for ATP, biosynthetic precursors, and reducing agents. Glutamine enters the cell through the amino acid transport protein called solute carrier family 1 member 5 (SLC1A5) and is converted into glutamate in the mitochondria via a dehydrogenation reaction catalyzed by GLS (Figure 1). Glutamate is converted by GLUD, alanine, or aspartate transaminase (TAs) to α-KG, which is an intermediate product of the TCA cycle [5]. In addition, the signal transduction molecules, viz., protein kinase B (PKB), RAS, and adenosine 5′-monophosphate (AMP)-activated protein kinase (AMPK) activate glycolytic enzymes and induce lactate production, requiring cancer cells to metabolize with glutamine to meet their increased energy needs [23]. Apart from energy supply, glutamine plays an important role in the biosynthesis of cancer cells. After glutamine enters the cell, ammonia can be produced by the action of GLS and GLUD, where the nitrogen element can be used directly in the synthesis of purines and pyrimidines [24]. It has also been found that glutamine produces α-KG [25], which is one of the raw materials for the synthesis of purines and pyrimidines, through a redox reaction to produce aspartic acid [26]. By promoting the synthesis of nucleotide precursors, glutamine catabolism in cancer cells facilitates the proliferation and division of cancer cells. Glutamine also represents a source of nitrogen in protein biosynthesis [24]; isotope tracing revealed that about half of non-essential amino acids required by tumor cells for protein synthesis are derived from glutamine [5]. Last but not least, under hypoxic conditions, tumor cells can also use glutamine to produce citric acid and fatty acids through reductive carboxylation reactions [27], while synthesizing dihydro-orotate acid to reduce the adverse effects of ammonia on tumor cells [28].

According to a recent study, Reinfeld and colleagues used positron emission tomography (PET) tracers to measure the acquisition and ingestion of glutamine by specific cell subpopulations in the tumor microenvironment; moreover, they found that myeloid cells are the ablest to absorb glucose from tumors in a series of cancer models. In contrast, cancer cells had the highest intake of glutamine [29,30]. They found that the significance of glutamine on tumor cells may be equally important. As with glycolysis, the breakdown of glutamine can also produce energy for cancer cells. Glutamine provides carbon and nitrogen to synthesize nucleotides such as glutathione, as well as other molecules, which are required for cancer-cell growth [23]. Many researchers have focused on this to understand if it inhibits the metabolism of glutamine in tumor cells and restrains the growth and proliferation of tumors. Searching for GLS or glutamine at https://clinicaltrials.gov/ (13 October 2019) collected seven relevant clinical trials, all related to GLS inhibitor CB-839. However, metabolic inhibitors are not very attractive in cancer treatment; part of the problem is that there are few ways to really limit it to malignant cells, which increases the likelihood of toxic effects on non-malignant cells [31]. Glutamine is the nutrient that transports carbon and nitrogen among the organs [5,32]; therefore, systemic disruption of their production and transport is expected to affect the other organs, such as the liver [33]. In addition, the use of metabolic inhibitors can also have an impact on immune cells such as T cells [34], and these effects are difficult to estimate. We know that CD8^+^ T cells have a significant effect in inhibiting tumor proliferation, metastasis, and prognosis [35], and inhibiting the activity of T cells is not our original intention. However, Leone et al. have recently reported that a new glutamine antagonist, JHU083, causes tumor regression in mice by preventing cancer cells from using the amino acid glutamine to feed anabolism [36]. It is worth noting that the drug not only inhibits the intake of cancer cells but also regulates the tumor environment, making it a more effective T-cell-friendly microenvironment, which in turn enhances their attack on tumors.

## 3. Effect of Metabolism on Immune Cells

Previous studies have shown that the immune system, e.g., CD8^+^ T cells, natural killer (NK) cells is a strong backbone for fighting against cancer [37,38]; at the same time, the body regulates new immune cells that also promote or inhibit tumor growth by altering the play of antitumor immunity. The relationship between the performance of these functions and cellular metabolism is inextricably linked. On the one hand, the body’s antitumor immunity requires metabolism to provide energy [39], while cellular metabolism can promote or inhibit the immune escape of tumor cells [40]. Finally, there is evidence that the metabolic characteristics of tumor cells themselves can influence the metabolism of immune cells [16]. Therefore, we need a deeper understanding of the metabolic mechanisms of immune cells in the tumor microenvironment (Figure 2).

### 3.1. TAMs

There is an abundant infiltration of macrophages in the tumor microenvironment, which are also known as TAMs [41]. Chemokines play a major role in chemotactic monocytes or macrophages outside the tumor into the tumor tissue. Chemokines such as colony-stimulating factor 1 (CSF1) and C-C motif chemokine ligand 2 (CCL2) secreted by tumor cells can recruit monocytes in peripheral circulation to the tumor microenvironment, and then monocytes differentiate into macrophages that are often classified as M1 and M2 types, with the former capable of killing tumor cells, while the latter favors tumor growth, metastasis, and weakening the effect of CD8^+^ T cells [42].

Glucose metabolism promotes macrophage differentiation toward tumor promotion. While the traditional view is that tumor cells produce lactate by glycolysis, recent findings suggest that macrophages are more capable of consuming glucose and producing lactate [29]. However, in any case, together, they create an acidic tumor microenvironment. The researchers found that lactate promotes the polarization of TAMs of the M2 type. Moreover, the polarized macrophages express more vascular endothelial growth factor (VEGF) and arginase-1 (ARG-1), the former of which promotes blood vessel growth, while the latter catalyzes the production of polyamines, which facilitate the proliferation of cancer cells [43]. At the same time, lactate inhibited the expression of macrophage ATP6V0d2, which was able to degrade hypoxia-inducible factor-2α (HIF-2α) via lysosomes [44]. In comparison, HIF shave been shown to play a far-reaching role in a variety of cancers [45]. In addition to facilitating tumor growth, M2 macrophages are also able to block antitumor immunity. Recent research investigations have confirmed that lactate can increase programmed death-ligand 1 (PD-L1) expression via nuclear factor kappa-light-chain-enhancer of activated B cells (NF-κB), which further promotes the metabolic activity and immunosuppression of macrophages [46].

The pro-tumor effect of TAMs also requires the support of amino acid metabolism. For example, methionine adenosyltransferase II alpha (MAT2A) increases the level of S-adenosylmethionine (SAM) in macrophages, and SAM promotes tumor-associated macrophage polarization through histone methylation [47]. Finally, previous studies found increased expression of several lipid metabolism genes in macrophages in tumor tissues [18], suggesting that TAMs cannot be performed without the regulation of lipid metabolism. Recent studies have found that reduced expression of monoacylglycerol lipase (MGLL) in TAMs facilitates tumor growth and that this molecule can catabolize 2-arachidonoylglycerol (2-AG), which promotes the cannabinoid receptor-2 (CB-2), leading to M2-type polarization of macrophages [48].

### 3.2. Regulatory T Cells (Tregs)

As a member of the CD4^+^ T cell family expressing forkhead box P3 (FOXP3), a cell is involved in suppressing the immune response in vivo and is capable of maintaining a homeostasis [49]. Tregs regulate the immune response to self and foreign antigens and prevent autoimmune disease [49]. However, the role of Tregs in suppressing the immune response in the tumor environment was not what we expected. Therefore, TGF-β secreted by Tregs is important for inhibiting the function of Tregs or immunosuppressive cytokines such as interleukin 10 (IL-10). In addition, it has been reported that Tregs can promote the polarization of TAMs to the M2-type, and thus indirectly weaken antitumor immunity [50]. The metabolic process of Tregs is the focus of researchers. In a recent review, Leone et al. found that although glucose uptake by Tregs is not high relative to that of effector T cells [11], glucose metabolism still plays an important role in Tregs’ function.

The use of the glycolysis inhibitor 2-deoxyglucose (2-DG) has been reported to decrease the expression of T-cell surface markers, such as cytotoxic T-lymphocyte-associated protein 4 (CTLA4), programmed cell death protein 1 (PD-1), CD71, and CD39 [51]. These molecules can inhibit the phosphorylation of the signal transducer and activator of transcription 5 (STAT5). The study also indicated that enolase-1, one of the enzymes involved in glycolysis, was able to specifically inhibit the expression of FOXP3 [51]. Interestingly, it was suggested by some previous researchers in their studies that in the tumor microenvironment, FOXP3 reduces glycolysis in Tregs by inhibiting MYC, and it in turn allows them to be free from lactate limitation [52]. In addition, Tregs being fundamentally different from CD8^+^ T cells and CD4^+^ T cells can utilize lactate in the tumor microenvironment through many metabolic enzymes, although they do not require lactate to survive [53]. Finally, effector Tregs have a stronger glycolytic capacity compared to central Tregs [54], which laterally verifies that glucose metabolism plays an important role in maintaining the immunosuppressive capacity of Tregs.

Like glycolysis, lipid metabolism can also promote the dominance of Tregs in tumor tissue. The researchers found that the use of ac-COA carboxylase inhibitors reduced the accumulation of fatty acids in tumor-associated Tregs, and decreased the proliferation of these cells, but the inhibitors did not show significant cytotoxic effects, suggesting that FAS may promote the proliferation of Tregs [55]. Moreover, the immunosuppressive ability of Tregs is supported by the reprogramming of fatty acid metabolism. Researchers propose that FAS-mediated de novo synthesis of fatty acids helps regulate T-cell expression of PD-1; this is attributed to the modulation of SREBP-cleavage-activating protein (SCAP)/sterol-regulatory-element-binding protein (SREBP) signal [56]. Meanwhile, gastric cancer cells with ras homolog family member A (RHOA) Y42 mutation produced free fatty acids in the microenvironment through phosphoinositide-3-kinase (PI3K)/PKB/mammalian target of rapamycin (mTOR) signaling, which also enhanced the immunosuppressive function of Tregs and promoted their aggregation in the low-glucose tumor microenvironment [57]. These studies addressed the role of targeted lipid metabolism on immune checkpoint inhibitors, which has important clinical implications. Of course, the specific mechanism of the effect of lipid metabolism on the regulation of T cells still needs to be further investigated.

### 3.3. Myeloid-Derived Suppressor Cells (MDSCs)

As a population of myeloid cells with immunomodulatory activity, MDSCs consist mainly of granulocytes and monocytes [58]. Unlike Tregs, this class of cells tends to appear in pathological states such as tumors, inflammation, and infections [59]. MDSCs play a crucial role in suppressing antitumor immune response in the microenvironment through the expression of ARG-1, as well as the secretion of cytokines, among other pathways [60].

Tumor glycolysis regulates the activation of MDSCs. Li et al., in a study of triple-negative breast cancer, suggested that glycolysis of tumor cells facilitates the growth of MDSCs and suppresses the body’s antitumor immunity, a phenomenon associated with the regulation of colony-stimulating factors (CSFs) by lactate dehydrogenase and autophagy [61]. Moreover, in a mouse model of pancreatic cancer, lactate, a product of glycolysis, was shown to activate MDSCs via the G protein-coupled receptor 81 (GPR81)/mTOR-HIF-1α/signal transducer and activator of STAT3 pathway [60]; this study simultaneously refreshed the understanding of radiation therapy for pancreatic cancer. As with tumor glycolysis, glutamine metabolism has a similar role. Glutamine metabolism maintains the stability of transcription factor liver activator protein (LAP), a molecule known to regulate the expression of granulocyte colony-stimulating factor (G-CSF), thereby influencing the convening of suppressor cells in the myeloid lineage around the microenvironment [62]. According to the results of this study, it is feasible to improve the therapeutic effect of anti-PD1 and anti-CTLA4 by blocking glutamine metabolism. Finally, the fatty acid metabolism of MDSCs can also influence their immunosuppressive function. MDSCs in the tumor microenvironment have high glycolytic activity [11]. It was demonstrated that inhibition of FAO in MDSCs reduced the rate of extracellular acidification in this cell type and reduced glycolytic activity in the cell [63]. This experiment also suggests that FAO maintains the expression of proteins such as ARG-1 and promotes the secretion of various CSFs, which favor the function of MDSCs.

### 3.4. CD4^+^ T Cells, CD8^+^ T Cells, and NK Cells

In the antitumor immune process, the function of cytotoxic T cells requires the complementary role of CD4^+^ T cells [64]. For example, CD4^+^ T cells induce dendritic cells to produce IL-12 and IL-15 to promote differentiation of cytotoxic T cells [35]. At the same time, CD4^+^ T cells themselves can produce cytokines to participate in the suppression of tumor cells, such as γ-interferon (IFN-γ), tumor necrosis factor-α (TNF-α), and IL-17 [65]. CD8^+^ T cells are the main driver of the body’s antitumor immunity, and they kill tumor cells through cytotoxic molecules such as perforin and granzyme or by inducing apoptosis in target cells [66]. NK cells have a role similar to that of CD8^+^ T cells, while they are also capable of killing tumor cells through antibody-dependent, cell-mediated cytotoxicity [67]. In addition, they are both capable of producing cytokines such as IFN-γ to suppress tumors [67,68]. These studies demonstrated the importance of CD4^+^ T cells, CD8^+^ T cells, and NK cells in the antitumor effect, and over the years, studies have continued to identify metabolic modulation of the tumor-suppressive function.

Glycolysis of cancer cells has a dramatic impact on CD8^+^ T cells and NK cells exerting antitumor immunity. It has been well demonstrated previously that tumor cells in the microenvironment consume glucose at an intense level (Figure 2) [16], creating an environment of glucose scarcity and large amounts of lactate, with glucose restriction impairing the antitumor effect of T cells [16]. Meanwhile, it was found that too high a concentration of lactate in the microenvironment was detrimental to the expression of nuclear factor of activated T cells (NFAT) in CD8^+^ T cells and NK cells, which ultimately reduced the manufacture of IFN-γ [69]. Glucose metabolism also has a positive effect on CD8^+^ T cells. In the literature, the research studies have suggested that reduced production of phosphoenolpyruvate (PEP), a glycolytic metabolite of CD8^+^ T cells and CD4^+^ T cells, was detrimental to the immune surveillance of T cells due to the competition for glucose by tumor cells, which may be related to the defective Ca ^2+^-NFAT signaling and T-lymphocyte activation [70]. This year’s paper shows that CD8^+^ T cells promote gluconeogenesis by upregulating phosphoenolpyruvate carboxykinase-1 (PCK-1), and that the G-6-P generated into the PPP generates NADPH that protects cells from ROS [71].

In addition, the effect of T-lymphocytes on tumors is also influenced by abnormalities of lipid metabolism in the tumor microenvironment. In 2020, it was reported that a high-fat diet reduced the number and function of cytotoxic T cells in a mouse model of colorectal cancer [72]. Firstly, researchers believe that abnormalities in lipid metabolism can damage T-lymphocytes or induce apoptosis in T-lymphocytes. For instance, it was reported that in mice with non-alcoholic fatty liver disease (NAFLD), the accumulation of free fatty acids disrupted the function of the mitochondrial electron transport chain of CD4^+^ T cells and generated more ROS to damage CD4^+^ T cells, thus promoting the growth of hepatocellular carcinoma [73]. Similarly, Ma et al. reported that CD36 was found to promote the uptake of arachidonic acid in CD8^+^ T cells, causing lipid peroxidation and ferroptosis, ultimately reducing antitumor immune function and sensitivity to immunotherapy [74]. However, Xu et al. found that CD36 could promote uptake of oxidized low-density lipoprotein by CD8^+^ T cells causing lipid peroxidation, thereby reducing the cytotoxic effects of such immune cells [75]. As a glycoprotein located on the cell membrane, CD36 expression in both TAMs and Tregs has been reported to promote tumor growth [76,77]; hence, CD36 can be considered an important target in cancer therapy. In addition, it has also been suggested that reprogramming of T-cell lipid metabolism reduces its antitumor effects by inhibiting glycolysis. In obese breast cancer mice, the transcription factor STAT3 was found to promote FAO in CD8^+^ T cells, thereby inhibiting glycolysis and reducing the tumor-suppressive effect of these cells [78]. To this end, the transcription factor STAT3 was activated by leptin in adipose tissue. In a nutshell, blocking lipid metabolic signaling in the tumor microenvironment can restore immune cell function and improve the sensitivity of immune checkpoint blockers to cancer, providing new ideas for future cancer therapy.

## 4. Metabolic Abnormalities in Cancer-Associated Fibroblasts (CAFs) and Their Significance

As an important component of stromal cells in the tumor microenvironment, CAFs mediate tumor cell proliferation, drug resistance, and immune escape by secreting various inflammatory ligands, growth factors, and extracellular matrix [79]. Firstly, metabolically heterogeneous CAFs can promote the growth and metastasis of cancer cells. Previous studies found enhanced glycolysis in some CAFs [80]. In recent years, researchers have found through various assays that colorectal CAFs exhibit more active FAS than normal fibroblasts; and these synthesized products can be taken up by colorectal cancer cells, thereby enhancing the metastasis of cancer cells [80]. Interestingly, metabolic abnormalities between CAFs and tumor cells can interact with each other. Yan et al. found that breast cancer cells can secrete miRNA-105 as exosomes (a small molecule that activates MYC signaling to enhance glycolysis and glutaminolysis in tumor-associated fibroblasts), producing nucleotides, α-KG, glutathione, and various amino acids that in turn support the growth of breast cancer cells [81]. Meanwhile, the researchers co-cultured fibroblasts and H1299 lung cancer cells, finding that the ROS level in H1299 cells increased, leading to an increase in the mRNA level of TGF-β in fibroblasts, which ultimately promoted the glycolysis level and lactate secretion. However, the oxidative phosphorylation function in H1299 lung cancer cells was enhanced [82]. Similarly, the researchers observed high glycolytic activity of CAFs in a subtype of pancreatic ductal adenocarcinoma by single-cell analysis; hence, the cancer cells exhibited a predominantly oxidative phosphorylation mode of function [83]. Importantly, this subtype showed a significant effect on treatment with immune checkpoint blockers. Of course, the mechanisms underlying the effects of abnormal CAFs metabolism on tumors still need to be further investigated, and these studies have an important role in the treatment and diagnosis of cancer.

## 5. Impact of Metabolism on Tumor Microenvironment

### 5.1. Differential Diagnosis of Cancer Using Metabolic Features

Taking advantage of the high glucose consumption of tumor tissue, molecules labeled with radionuclides (^18^F-FDG) are injected as probes to locate tumors in the body; the combination of CT and PET has since increased the advantages of PET [84]. Today, PET is widely used in clinical practice, with various data reporting its advantages in the diagnosis of cancer [85,86,87]. In addition to the differential diagnosis of tumors, PET-CT can also be used to assess the effectiveness of immunotherapy, especially for immune checkpoint inhibitors such as CTLA4 and PD-1 [88]. However, the distribution of tumor molecular targets cannot be observed by using the ^18^F-FDG-PET. At the same time, the use of ^18^F-FDG as an imaging agent is still limited due to high glucose depletion of immune cells causing false positives in PET-CT for tumor diagnosis [29,89]. In addition to ^18^F-FDG, more imaging techniques and markers are being explored that could help form potential scenarios for precision diagnosis in the future. ^11^C-DASA-23, which can efficiently activate pyruvate kinase M2 (PKM2), can be used as an imaging agent to efficiently display gliomas in the cranial cavity of mice [90]. At the same time, ^18^F-DASA-23 can achieve a similar effect [91]; this is something that ^18^F-FDG-PET cannot do. Similar to the metabolism of glucose, the metabolic characteristics of glutamine can also be used to design suitable imaging agents. For example, through clinical trials, researchers found that ^18^F-FGln showed better affinity as a visualizing agent in cancers such as glioma and breast cancer [92]; this study demonstrated the potential use of this imaging agent for the screening and prognosis of some cancers as well as to compensate for the lack of ^18^F-FDG in specific cases.

### 5.2. Targeting Metabolism to Improve the Effectiveness of Cancer Immunotherapy

In recent years, the advent of immune checkpoint inhibitors has changed the landscape of the human fight against tumors [93]. However, their efficacy is still limited—a significant proportion of patients are not sensitive to immune checkpoint therapy [94,95], such as the use of chemotherapy drugs [96]. Therefore, the use of metabolic modulation of antitumor immunity can help to improve the sensitivity of immunotherapy.

Since a microenvironment with high lactate is not conducive to antitumor immunity and promotes tumor growth [97], targeting lactate production can enhance the effect of immunotherapy. Firstly, the simplest idea was proposed, namely, to directly neutralize lactate in the microenvironment with alkaline drugs, which were found to significantly enhance the effect of immune checkpoint inhibitors [98]. At the same time, in the cellular and animal models of Renner et al., monocarboxylate transporter (MCT) inhibitors significantly enhanced the therapeutic effect of PD-1 and CTLA4 inhibitors [99]. 

Regarding glutamine metabolism, researchers have also developed many drugs that have shown preliminary results in animal studies. The priority was to inhibit glutamine uptake by tumor cells. For example, according to Byun et al., the combination of glutamine transporter inhibitors and PD-L1 blockers significantly enhanced the therapeutic effect of immune checkpoint blockers in lung and colon cancers [100]. At the same time, they clarified that PD-L1 expression limited glutamine utilization by T-lymphocytes. Similarly, some drugs target GLS to block intracellular glutamine utilization, such as JHU083 and CB839. In mice with colon cancer, the PD-1 blocker alone showed almost no response, while the combination with the GLS inhibitor JHU083 resulted in a nearly 100% increase in antitumor effect [36]. Finally, the experimental GLS inhibitor CB839 was found to effectively promote the aggregation of CAR-T in tumor tissues [10]. In this review, Table 1 summarizes the mechanisms of these drugs and their impact on immunotherapy for cancer in previous studies. These might be powerful weapons to improve the efficacy of immunotherapy for solid tumors. In conclusion, the current research focused on the metabolism and immunity in oncology is flourishing; and the altered metabolism has been shown to enhance the therapeutic effect of immunotherapy on solid tumors in some experiments, with more efficient drugs with low side effects being due to be developed in the future.

## 6. Discussion

We elucidated the impact and specific mechanisms of various metabolism pathways involved in the tumor microenvironment. In tumor cells, glycolysis mainly provides rapid energy and the opportunity to survive in a microenvironment with low oxygen as well as a lack of nutrients, while glutamine mainly provides raw materials for the synthesis of amino acids and nucleotides, as well as fatty acids. The competition for glucose and the massive release of lactate by tumor cells shape an immunosuppressive microenvironment, such as promoting the differentiation of macrophages to the M2-like macrophages, as well as recruiting more MDSCs. Thus, limiting the function of lymphocytes and NK cells is one of the important reasons for the poor effect of immunotherapy in solid tumors; meanwhile, the consumption of glucose by macrophages deserves our attention. In addition, the accumulation of fatty acids affects a variety of cells in the microenvironment, but the exact mechanism of this phenomenon needs to be further investigated. Finally, we have listed the progress of metabolic co-immunotherapy in some animal experiments (Table 1). We believe that immunotherapy has great potential in solid tumors in the future with the development of drug research.

## Figures and Tables

**Figure 1 biomolecules-12-00580-f001:**
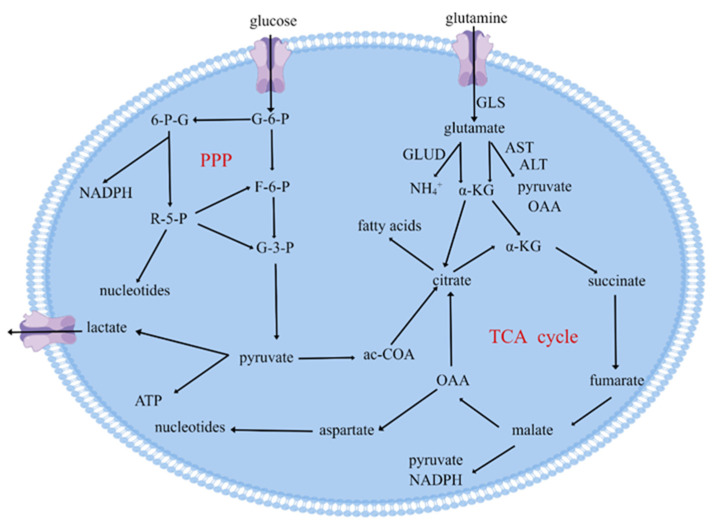
Metabolism provides superior conditions for the growth and proliferation of tumor cells. According to the Warburg effect, glucose tends to be eventually oxidized to lactate in tumor cells instead of acetyl coenzyme A (ac-COA) while providing large amounts of ATP rapidly. During the progress of glycolysis, G-6-P can change into 6-phosphate–gluconate (6-P-G) and R-5-P, which usually appear in PPP. PPP can synthesize fatty acids, nucleotides, NADPH, and other products to promote tumor growth and division, among which NADPH can protect cancer cells from the damage of ROS. The growth of cancer cells also requires the breakdown of glutamine. Glutamine entering cells is converted to glutamate by GLS, which is catalyzed by glutamate dehydrogenase (GLUD) and TAs to α-ketoglutarate (α-KG), amino acids, and ammonium salts. Glutamate pyruvate aminotransferase (ALT) and aspartate aminotransferase (AST) can also catalyze the generation of α-KG and oxaloacetate (OAA) from glutamate. α-KG enters the TCA cycle to generate energy. α-KG can also produce citric acid directly through reductive carboxylation and eventually synthesize fatty acids.

**Figure 2 biomolecules-12-00580-f002:**
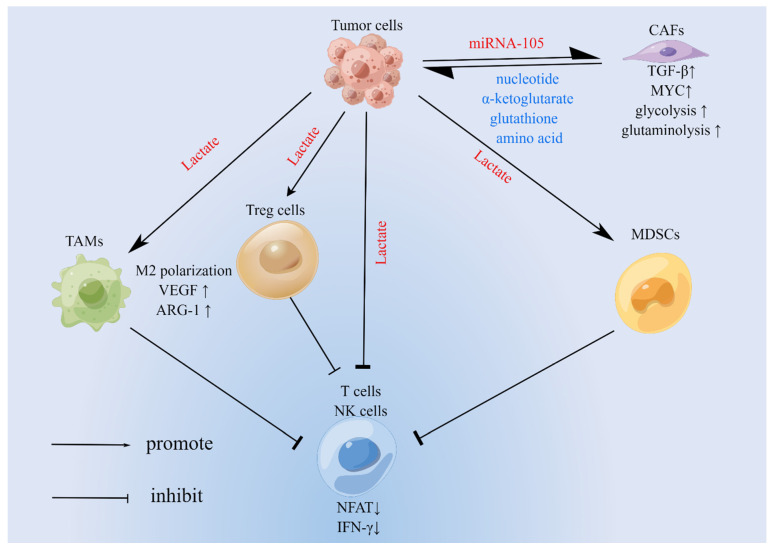
Interaction of metabolism in the tumor microenvironment. As we can see from the figure, tumor cells and TAMs are the main members of lactate production in the tumor microenvironment. The accumulation of lactate in the microenvironment stimulates the activation of TAMs to the M2 phenotype and the activation of more MDSCs, in addition to the ability of Treg to utilize lactate. These cells inhibit T-lymphocytes and NK cells in the microenvironment, which is detrimental to their recognition and destruction of tumor cells.

**Table 1 biomolecules-12-00580-t001:** Some drugs for targeted metabolism combined with immunotherapy.

Drug	Targeted Metabolism	Mechanism	Appropriate Immunotherapy	Source
Diclofenac	Glycolysis	Inhibit lactate transporter protein	Anti-PD-1 treatment	[99]
Bicarbonate	Glycolysis	Directly increase pH value	Anti-CTLA4 treatment Anti-PD-1 treatment	[98]
JHU083	Glutaminolysis	Inhibit GLS activity	Anti-PD-1 treatment	[36]
V-9302	Glutaminolysis	Inhibit glutamine transporter protein	Anti-PD-L1 treatment	[100]
CB839	Glutaminolysis	Inhibit GLS activity	CAR-T cell therapy	[10]

## Data Availability

Not applicable.

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
