# Peer review of "Effects of Glucose Metabolism, Lipid Metabolism, and Glutamine Metabolism on Tumor Microenvironment and Clinical Implications"

_biomolecules, 2022, doi:10.3390/biom12040580_

Round 1

Reviewer 1 Report

This review discussed some typical metabolic features of tumor cells and the infiltrated immune cells, but not enough in-depth. The tumor microenvironment does not only include tumor cells and the immune cells, but also other types of stromal cells, for example, cancer-associated fibroblasts. On the other hand, other types of metabolism, such as lipid metabolism is barely mentioned. And also, the connections between different types of metabolism need to be better indicated. For example, the connection between glycolysis and PPP in glucose metabolism through GAPDH. 

Author Response

We have given some revision according to your suggestion.

  1. Please see the line366-392 in page 9, we have added some discussion about the cancer-associated fibroblasts.
  2. Please see the line 257-273 in page 6, line 334-357 in page 7, we have added some discussion about the lipid metabolism.
  3. Please see the line93-96 in page 2, We have added that the connection between glycolysis and the PPP is through glucose-6-phosphate dehydrogenase (G-6-PDH).

All of the revisions have been marked by the high light. Please check them.

Reviewer 2 Report

This generally well written manuscript gives a good summary of the impact of glucose and glutamine metabolism on cancer and especially the various celltypes involved in tumor growth. The presentation and line of argumentation is very systematic and easy to follow, including the two very catchy schematic figures.

While the english is generally good and very readable, a careful editorial check including thourogh proof-reading and in several instances correction of typing, syntax and sentence structure is needed. This is expecially apparent in the abstract (line e. g. lines 9, 10, 14) and spreads though the whole manuscript.

Author Response

Thank you very much for your suggestion.We have given some revision according to your suggestion as following:

1.Please see the line 10,11,15 in page 1, we have corrected the illustrations and added notes for the abbreviations.

2.Please see the other highlighted parts, we have fixed the grammar, spelling and other errors.

We uploaded two versions, one is the revisions with track changes, and another is the revisions marked by the high light. Please check them.

If any problem, please let me know.

Hoping for your reply.

Reviewer 3 Report

In this study, authors describe the effects of metabolism on tumor cells and effector cells in the tumor microenvironment 

Manuscript needs minor revisions:

- Some typing errors are present in the manuscript (e.g. Introduction section, line 51, "Broadly 50 speaking, Inhibition of one of [...]": inhibition is written in upper case, but should be in lower case). Please, check the whole text.  - Moderate English editing is required. I suggest to contact a native speaker. - Manuscript title selected by authors is "Impact of metabolism on the tumor microenvironment and its clinical significance". However, authors describe just two types of metabolism: glucose metabolism and glutamine catabolism. Thus, I suggest to change the current title with another more specific, or to add some paragraphs related to other metabolism involved in this field. In this regard, it has been recently demonstrated that iron metabolism is strongly connected with the tumor microenvironment and how this affects both tumor-associated macrophages and tumor-infiltrating lymphocytes functions [DOI: 10.3390/cells10020303].

Author Response

Thank you very much for your suggestion.We have given some revision according to your suggestion as following:

1. Please see the line 3-4 in page 1, we have revised the title to make it more specific.

2. Please see the line 54 in page 2, we have revised the case of “inhibition”.

3. Please see the other highlighted parts, we have fixed the grammar, spelling and other errors.

Please check them.

If any problem, please let me know.

Hoping for your reply.

Round 2

Reviewer 1 Report

I think it is good to publish.